# Assessing the pattern of key factors on women's empowerment in Bangladesh: Evidence from Bangladesh Demographic and Health Survey, 2007 to 2017–18

**Sahera Akter** [ORCID] *, **Md. Solayman Hosen** [⊙], **Md. Shehab Khan** [⊙], **Bikash Pal** [ORCID]

Department of Statistics, University of Dhaka, Dhaka, Bangladesh

⊙ These authors contributed equally to this work.
* sahera.akter0102@gmail.com

## Abstract

### Background

With half a female population, empowering women can be a key factor in our country's global advancement. Focusing on household decision-making and attitudes toward wife beating, our study addresses the dearth of research exploring how different socio-economic and demographic factors associated with women's empowerment evolve over the past decade in Bangladesh (from BDHS 2007 to BDHS 2017–18).

### Methods

Data from four waves of Bangladesh Demographic and Health Survey (BDHS, 2007 to BDHS, 2017–18) were used in this study. We put forth two domains—household decision-making and attitudes toward domestic violence—to assess women's empowerment. Principal component analysis (PCA) was employed to create women's empowerment index. To assess the unadjusted association between the selected covariates and women's empowerment, Pearson Chi-square test and ANOVA F test have been used, while adjusted association has been analyzed through proportional odds model (POM).

### Results

In BDHS 2017–18, women from urban areas experienced 'high' empowerment than women in rural areas (56.08% vs. 45.69%). A notable change has been observed in the distribution of women's empowerment index by education over the survey years. Findings also showed that in all the survey years, division, place of residence, education level, number of living children, media exposure, wealth index, working status, and relationship with household head have been found to have significant association with women's empowerment index. For instance, women who completed secondary education in 2007, 2011, 2014, and 2017–18, respectively have 14.4%, 31.8%, 24.6%, and 39.6% higher odds of having empowerment compared to those who were uneducated. Further, age at first marriage, spousal age gap, NGO membership etc. emerged as a contributing factor in specific survey years.

**Data Availability Statement:** The datasets utilized in this study are available for access through the DHS repository. Individuals interested in obtaining

the data can do so by submitting a formal request via the DHS website at the following link: https://dhsprogram.com/data/available-datasets.cfm.

**Funding:** The author(s) received no specific funding for this work.

**Competing interests:** The authors have declared that no competing interests exist.

## Conclusion

Our study affirmed that, over a ten-year period, women were more likely to protest against physical violence and to participate in various decision-making regarding their personal and social life. Empowerment is notably higher among women in urban residents, those with secondary education, 1–2 children, media exposure, and employment. Policy recommendations should emphasize targeted measures to raise awareness and empower uneducated, unemployed, economically disadvantaged, and physically oppressed women.

## Introduction

Women's empowerment is a pivotal focus in global development, recognized as essential for sustainable economic growth and poverty reduction [1]. Defined as an interactive process facilitating personal and communal change, empowerment enables individuals to impact institutions affecting their lives and communities [2]. Specifically, women's empowerment denotes their full participation in economic, political, and social spheres [2,3], critical for comprehensive social and economic development.

In developing countries, gender inequality poses a significant challenge [4], violating human rights and impeding both women's empowerment and national development. The Global Gender Gap Index, incorporating educational attainment, health, political empowerment, and economic participation, highlights Bangladesh's standing as the 65th country out of 156, indicating persistent gender inequality [5]. Notably, Bangladesh excels in women's political empowerment, with women well-represented in national legislatures [6]. Despite these strides, challenges persist, as evidenced by the prevalence of gender bias and violence against women, particularly in less-developed countries.

Aligned with international commitments such as the Convention on the Elimination of All Forms of Discrimination against Women (CEDAW) and the Sustainable Development Goals (SDGs), Bangladesh has made notable efforts to eradicate gender-based discrimination and violence [7–9]. The importance of women's empowerment is underscored by the global Sustainable Development Goal 5, aiming to end gender imbalance and violence against females by 2030 [10]. Additionally, the third Millennium Development Goal emphasizes progress toward closing the gender gap and enhancing women's political and economic influence [8].

In Bangladesh, despite these commitments, challenges persist. Gender-based violence, especially within households, remains a concern, affecting approximately 54.2% of married women [11]. The sheer size of the female population, constituting 50.46% of the total, accentuates the potential for substantial societal progress through increased women's empowerment [12].

Numerous studies have explored potential variables influencing women's empowerment in Bangladesh and globally [13–17]. For instance, studies using data from DHS 2010 and BDHS 2014 proposed indicators like women's participation in household decision-making and opinions concerning wife abuse [13,14]. Further, a study emphasized primary components of women's empowerment, including employment status, self-worth, self-confidence, decision-making ability, and awareness [15]. An evidence-based analysis suggested indicators such as personal freedom, household decision-making, domestic financial decisions, and political independence, exploring the impact of social media, education, community cultural values, women's employment, and household participation rate on women's empowerment [16]. Another study focused on women's empowerment through access to health information, finding that urban-dwelling, educated, working, middle-aged women had better decision-making abilities [17].

This study aims to identify factors contributing to women's empowerment in Bangladesh, filling a gap in existing literature. Unlike prior studies, our research uniquely utilizes four waves of BDHS data (2007, 2011, 2014, and 2017–18) to assess temporal changes in covariates impacting women's empowerment. Focusing on household decision-making and attitudes toward wife beating, our study addresses the dearth of research exploring how these factors evolve over the past decade in Bangladesh. Through a comprehensive analysis of evolving factors, we strive to contribute valuable insights into the dynamic landscape of women's empowerment in Bangladesh.

## Data and methodology

### Data

Data from the nationally representative Bangladesh Demographic and Health Survey (BDHS) have been extracted for conducting analysis on women's empowerment. To explore the changes in women's empowerment for one decade, we have used data from four waves of BDHS, collected in 2007, 2011, 2014, and 2017–18. These were nationally representative cross-sectional surveys based on a two-stage stratified sample of households. The details of the survey design are described in detail elsewhere [18–22].

In general, in the first stage, a certain number of enumeration areas, EAs (EAs varied for different BDHS) were selected with probability proportional to EA size and with independent selection in each sampling stratum. In the second stage of selection, a fixed number—30 households per cluster—were selected with an equal probability of systematic selection from the newly created household listing. In these surveys, only ever-married women of age 15–49 were interviewed, an exception was found for BDHS 2011 in which ever-married women aged 12–49 were interviewed. But making it analogous to other datasets, we have only taken information about the women who belong to reproductive age 15–49. Missing observations have been eliminated, leaving us with a total of 8931, 16274, 16350, and 18723 observations, respectively.

### Ethics approval and consent to participate

The Institutional Review Board of ICF International, Rockville, Maryland, USA (Macro International is now known as ICF International) reviewed and gave their approval to The Demographic and Health Surveys (DHS) Program. 2007, 2011, 2014, and 2017–18 Bangladesh DHS were categorized under that approval. Furthermore, the 2011 and 2017–18 Bangladesh DHS also received approval from another ethical committee: The Bangladesh Medical Research Council. These BDHS were implemented under the authority of the National Institute of Population Research and Training (NIPORT) of the Government of the People's Republic of Bangladesh with financial support from USAID/Bangladesh. Prior to asking, informed consent was obtained from each survey participant. Respondents who refused to consent were not included in the survey.

### Outcome variable

**Construction of women's empowerment indices.** In the conceptual framework (Fig 1), we considered a total of nine questions which were classified into two broad dimensions; household decision-making and attitudes toward wife beating [19–24]. BDHS collected information on household decision-making: (1) their own health care, (2) major household purchases, (3) their child's health care (this information is missing in BDHS 2017–18 data), and (4) visits to their family or relatives. For the above decisions, we made four binary variables

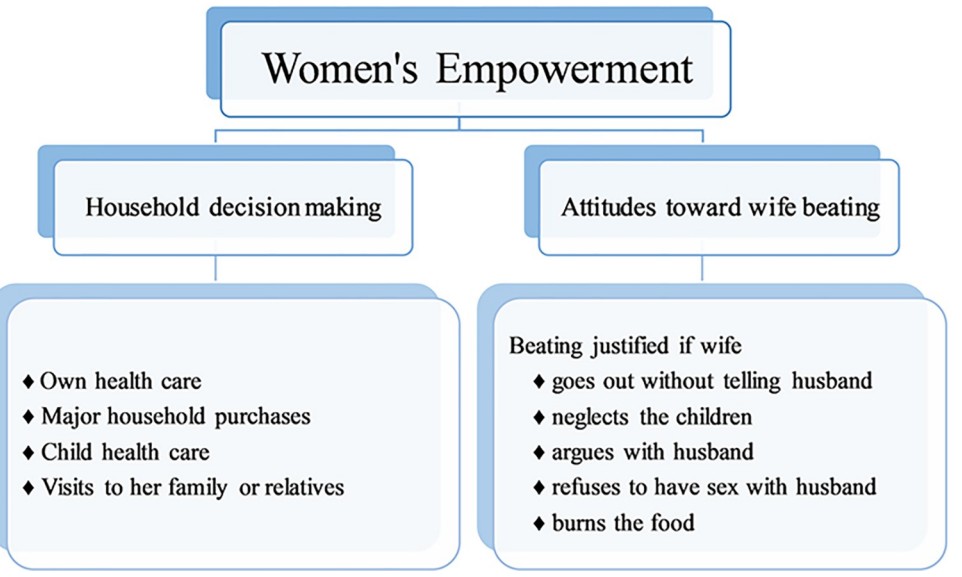

**Fig 1. Conceptual framework for women's empowerment.**

(yes/no) where "yes" was defined for the responses of "respondent alone", "respondent and husband/partner" or "respondent and other person", and "no" for otherwise. This index is positively related to women's empowerment and for analysis purposes, these questions were recoded (i.e., 1 for "yes" and 0 for "no"). The second index, containing five indicators, was used to gauge women's attitudes toward wife beating. Women were asked if they thought a husband had the right to hit his wife if she (1) refused to have sex with him, (2) left the house without his consent, (3) neglected the children, (4) argued with him, or (5) burned the food. The response "yes" to the above questions means that she believes wife-beating is justified and "no" means she rejects wife-beating for that particular reason. As this index is negatively related to women's empowerment, for the analysis purposes, these questions were recoded oppositely from the first indicator (i.e., 1 for "no" & 0 for "yes"). For creating the women's empowerment index (WEI), principal component analysis (PCA) has been used with all nine indicators (4 decisions and 5 reasons) [14,18]. Principal components (PCs), with their unique variance features, are normalized linear combinations of random (original) variables. Variance of a principal component is the eigenvalue associated with it. Ratio of $k^{th}$ eigenvalue to the sum of all the eigenvalues represents the proportion of variation explained by the $k^{th}$ component [25]. Here, in this study, after PCA, nine PCs (for BDHS 2007 and 2017–18, eight PCs) have been generated among which the first principal component was regarded as the women's empowerment score (WES) since most of the variations in the data were explained by it. The WES was further broken down into 3 equal portions based on quantiles; labeled low, middle, and high for domains below the first, in between the first and second, and above the second quantile, respectively. Finally, our preferred outcome metric is the score index with three ordered categories (i.e., low, medium, and high), where these categories indicate order-wise how empowered a woman is (i.e., low means women have low empowerment) [14,18].

## Covariates

On the basis of some previous works of literature, the covariates included in this study are age at first marriage, spousal age gap, respondent's education level, education gap, respondent's current working status, number of living children, religion, number of household members,

division, place of residence, media exposure, NGO membership, wealth index, husband's occupation, relationship with household head, and sex of household head [14,26–30]. All of these variables could not be directly extracted from the survey. We had to construct many of these with the help of directly available variables. The definition and their measurements (including categories) of variables are given in Table 1.

## Statistical analyses

Simple descriptive, bivariate, and multivariate statistical analyses were performed to achieve different objectives in this study. The univariate analysis was performed to investigate the

**Table 1. Definition and measurements of the variables based on BDHS data (from 2007 to 2017–18).**

| Covariates | Measurements |
|---|---|
| Division | Original dataset had eight divisions: Barisal, Chittagong, Dhaka, Khulna, Mymensingh (missing in 2007, 2011, and 2014 BDHS data as then it was not declared as a division), Rajshahi, Rangpur (was not declared as a division at the time of 2007 BDHS), Sylhet [19–22]. |
| Place of residence | This variable was categorized as: urban and rural [19–22]. |
| Age at first marriage | Age at first marriage was categorized into two categories as: below 18 years and 18 or 18+ years [14,27]. |
| Spousal age gap | Spousal age gap was measured by taking the difference between spousal ages [14]. |
| Respondent's education level | Four categories were created by the survey as: no education, primary, secondary, and higher [14,19–22]. |
| Education gap | The difference between the education levels of the spouses was divided into three categories—no gap, wife with lower education, and wife with higher education [14,27]. |
| Number of living children | Number of living children was categorized into four categories: no child, 1–2, 3–4, and 4+ children [19–22,27]. |
| Difference between sons and daughters | This variable was made by deducting the total number of daughters from the total number of sons. Negative results indicate a gap in which there are more daughters, positive numbers indicate more sons, and zero means simply no difference between daughter and son [14]. |
| Number of household members | Categorized into three categories: 1–4 members, 5–7, and 7+ members [19–22]. |
| Wealth index | The original dataset had five categories: poorest, poorer, middle, richer, and richest [19–22]. |
| Religion | Islam, Hinduism, Buddhism, and Christianity comprised the four categories of religion in the original dataset. However, for the sake of the study, the religion has been reclassified into two groups: Muslim (Islam) and non-Muslim (Hinduism, Buddhism, and Christianity) [18,19]. |
| Media exposure | A woman is exposed to the media if she has access to either of the media sources (Categories: Yes/No) based on the variables of watching TV, listening to the radio, and reading newspapers and magazines [14,26,28–30]. |
| NGO membership | Women's NGO membership was divided into two groups: "yes" if they are members of any one of the following organizations: Grameen Bank, BRAC, BRDB, ASHA, Proshika, Mother's Club, or other NGOs; and "no" if they are not [14,28,29]. However, because NGO activities are considerably declining in Bangladesh, data on these aspects was missing from the most recent BDHS 2017–18 data. The fact that the government offers more facilities than NGOs could be one of the causes [30]. |
| Working status | Working status had two categories: yes and no [14,19–22,26]. |
| Husband's occupation | Husband's occupation was categorized into seven categories as farmer, labor, service, large business, small business, unemployed and other [14]. |
| Relationship with household head | Survey created eleven categories as: head, wife, daughter, daughter-in-law, grand-daughter, mother, mother-in-law, sister, other relative, adopted/foster child and other [19–22]. |
| Sex of household head | Sex of household head was divided into two groups: male and female [19–22]. |

individual frequency percentage of the selected two indicators comprising women's empowerment in the four consecutive BDHS surveys. In the bivariate analysis of women's empowerment with different selected variables, the Pearson Chi-square test and ANOVA F-test have been applied as an attempt to find out the unadjusted association between women's empowerment and selected covariates. The covariates with p-value <0.05 has been considered significantly associated with the outcome variable and obtained significant covariates in at least any one study year have been included in the multivariable regression model. Since we have an ordinal response variable, the ordinal logistic regression model was used to find out the adjusted effects of covariates on women's empowerment. To assess the goodness of fit for the fitted model, the likelihood ratio test (LRT) has also been used [26]. All the analyses were performed with the help of two statistical software packages: SPSS version 20, for data sorting, cleaning, etc. and STATA version 14.0, for the analysis purpose.

**Ordinal logistic regression model.** With qualitative response variables, binary logistic, ordinal logistic, and multinomial logistic regression models can be used for analyzing data. But when analyzing polychotomous data, the ordinal logistic regression model gives a more accurate and efficient estimate of regression coefficients where the response variable acts in an ordinal way with each predictor. In our study, we have response variable (women's empowerment) with three ordered categories (low, medium, and high). Hence, proportional odds model (POM) has been employed in this study.

Suppose a response variable $Y$ with categorical outcomes, denoted by 0, 1, 2,. . .,$k$, and let $\tilde{x}$ denote a $p$- dimensional vector of covariates.

If $Pr[Y \geq j|\tilde{x}] = \pi_j$ is the cumulative logistic distribution function,

Then,

$$odds = \frac{Pr[Y \geq j|\tilde{x}]}{1 - Pr[Y \geq j|\tilde{x}]} = \frac{\pi_j}{1 - \pi_j} = \exp\left(\alpha_j + \tilde{x}'\tilde{\beta}\right) \tag{1}$$

And the dependence of $Y$ on $\tilde{x}$ for the proportional odds model has the following representation:

$$logit\left(\pi_j\right) = log\left[\frac{\pi_j}{1 - \pi_j}\right] = \alpha_j + \tilde{x}'\tilde{\beta} = \alpha_j + \beta_1 x_1 + \beta_2 x_2 + \cdots + \beta_p x_p, \ j = 1, 2, \ldots, k \& i = 1, 2, \ldots, p \tag{2}$$

where, $\alpha_j$ is the $j^{th}$ intercept, $\tilde{x} = (x_1, x_2, \ldots, x_p)\prime$ and $\tilde{\beta} = (\beta_1, \beta_2, \ldots, \beta_p)\prime$ is the $p \times 1$ vector of unknown regression coefficients corresponding to $\tilde{x}$ [31–34].

## Results

To see how the situation involving women's empowerment improved over the study years, the percentage distribution of variables regarding decision-making and attitudes toward violence has been utilized (Fig 2). In BDHS 2007, 86.8% of women reported having freedom of decision-making, which decreased in 2011 to 82.2%. But in 2014, it again increased about 0.9%, and hit a record high of 88.1% in 2017–18. This implies that women in Bangladesh have acquired greater autonomy in household decision-making. Conversely, the percentage of women who were against violence towards them increased about 28.9% from 2007 to an all-time high percentage of 97.7% in 2011. In the next survey year, it dropped to 71.1%. However, in 2017–18, it increased to 80.4%. While there was a spike in 2011, women's perceptions of the justification of violence have improved across the survey years.

Table 2 outlines the percentage distribution of the women's empowerment index (WEI) by selected covariates over the survey years. It reveals that women living in urban areas

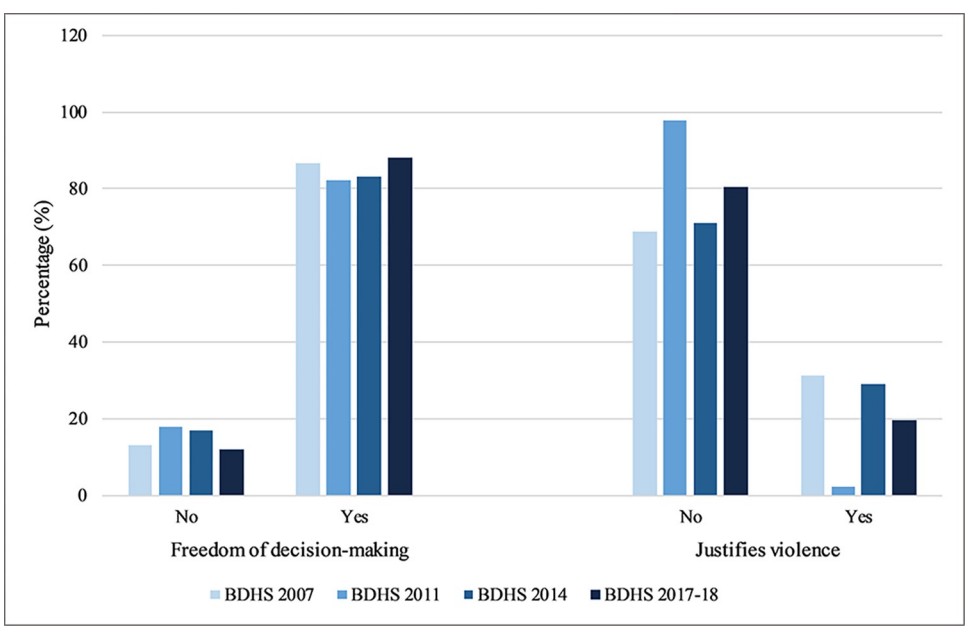

**Fig 2. Distribution of freedom of decision-making and attitudes toward justification of violence over four waves of BDHS (2007 to 2017–18).**

experienced 'high' empowerment than their counterparts in rural areas in four waves of BDHS. For instance, urban (vs. rural) women who reported a high level of empowerment were 39.56% (vs. 29.01%), 36.90% (vs. 26.89%), 36.76% (vs. 29.12%), and 56.08% (vs. 45.69%), respectively, from 2007 survey to 2017–18 survey. Over the survey years, a notable change has been observed in the distribution of WEI by education. To be specific, the highest percentage of women with secondary education getting 'high' empowered was recorded in 2017–18 BDHS

(48.05%), followed by 2007 BDHS (33.32%), 2014 BDHS (31.53%), and 2011 BDHS (30.29%). In addition, higher-educated women reported having 'high' empowerment were 3.5 times as having 'low' empowerment (15.81% against 56.05%) in BDHS 2017–18. Moreover, the level of empowerment among women with 1–2 children is higher than that of women with no children. In 2017–18 BDHS, 50.76% of women having 1–2 children experienced high empowerment, which was 34.07% according to BDHS 2007. Similarly, compared to other study years, a greater percentage of women who are exposed to mass media (51.26%) reported having high empowerment in 2017–18 BDHS. However, from the Pearson Chi-square and ANOVA F-test p-values, the study finds significant bivariate association for all the selected covariates with women's empowerment.

A summary of the multivariable analysis results that were obtained by fitting the proportional odds model is compiled in Table 3. The overall goodness of fit of the model for each study survey has been checked through the likelihood ratio test (LRT) based on chi-square distribution. As the p-value for the corresponding LRT was less than 0.001 for each survey model, the LRT ensured that model was well fitted. Multivariable analysis results demonstrate the adjusted odds ratios (OR) for women's empowerment from BDHS 2007 to 2017–18. All the covariates except the sex of the household head were found to have a significant effect on women's empowerment in at least any one study year. The odds of experiencing empowerment

differed significantly in all the survey years by division, place of residence, education level, number of living children, media exposure, wealth index, working status, and relationship

**Table 2. Women's empowerment index (WEI) by demographic and socioeconomic characteristics, BDHS 2007 to BDHS 2017–18.**

| Covariates | BDHS 2007 | | | BDHS 2011 | | | BDHS 2014 | | | BDHS 2017–18 | | |
|---|---|---|---|---|---|---|---|---|---|---|---|---|
| | Low | Medium | High | Low | Medium | High | Low | Medium | High | Low | Medium | High |
| **Division** | | | | | | | | | | | | |
| Dhaka | 26.36% | 36.26% | 37.37% | 29.24% | 37.21% | 33.55% | 26.14% | 34.21% | 39.65% | 21.42% | 25.74% | 52.84% |
| Chittagong | 39.81% | 30.53% | 29.66% | 37.88% | 34.52% | 27.61% | 33.86% | 31.70% | 34.43% | 24.60% | 26.20% | 49.20% |
| Barisal | 41.86% | 34.29% | 23.85% | 32.17% | 33.84% | 33.99% | 40.82% | 32.60% | 26.58% | 29.32% | 26.86% | 43.83% |
| Khulna | 27.50% | 35.12% | 37.37% | 26.74% | 41.36% | 31.91% | 36.30% | 33.77% | 29.93% | 23.60% | 27.26% | 49.15% |
| Mymensingh | - | - | - | - | - | - | - | - | - | 20.75% | 20.02% | 59.23% |
| Rajshahi | 26.12% | 38.19% | 35.68% | 40.66% | 37.11% | 22.23% | 33.12% | 38.91% | 27.97% | 24.87% | 25.78% | 49.36% |
| Rangpur | - | - | - | 27.86% | 32.91% | 39.24% | 29.71% | 37.83% | 32.46% | 22.66% | 27.51% | 49.83% |
| Sylhet | 46.61% | 22.71% | 30.68% | 40.43% | 36.20% | 23.37% | 41.09% | 31.54% | 27.37% | 27.16% | 31.78% | 41.06% |
| p-value | <0.001 | | | <0.001 | | | <0.001 | | | <0.000 | | |
| **Place of residence** | | | | | | | | | | | | |
| Urban | 27.75% | 32.69% | 39.56% | 26.07% | 37.03% | 36.90% | 27.96% | 35.28% | 36.76% | 19.58% | 24.35% | 56.08% |
| Rural | 37.22% | 33.77% | 29.01% | 37.24% | 35.87% | 26.89% | 36.90% | 33.98% | 29.12% | 26.77% | 27.54% | 45.69% |
| p-value | <0.001 | | | <0.001 | | | <0.001 | | | <0.001 | | |
| **Age at first marriage** | | | | | | | | | | | | |
| Age<18 | 35.23% | 33.48% | 31.28% | 34.46% | 35.92% | 29.62% | 35.16% | 34.35% | 30.49% | 25.46% | 25.94% | 48.61% |
| Age≥18 | 26.92% | 32.85% | 40.23% | 29.53% | 37.48% | 32.99% | 29.69% | 34.66% | 35.65% | 20.50% | 27.59% | 51.91% |
| p-value | <0.001 | | | <0.001 | | | <0.001 | | | <0.001 | | |
| **Spousal age gap** | | | | | | | | | | | | |
| Mean | 9.79 | 9.34 | 9.38 | 9.34 | 9.03 | 9.13 | 9.19 | 8.79 | 8.91 | 8.44 | 8.36 | 8.45 |
| (SD) | (7.43) | (6.96) | (7.05) | (5.74) | (5.42) | (5.31) | (6.04) | (5.46) | (5.46) | (5.35) | (5.21) | (5.36) |
| p-value[A] | 0.001 | | | <0.001 | | | <0.001 | | | 0.055 | | |
| **Education level** | | | | | | | | | | | | |
| No education | 37.08% | 33.27% | 29.64% | 37.10% | 35.50% | 27.40% | 37.82% | 31.17% | 31.01% | 27.66% | 23.14% | 49.20% |
| Primary | 37.10% | 32.75% | 30.15% | 37.14% | 34.29% | 28.57% | 36.51% | 34.32% | 29.17% | 26.27% | 25.37% | 48.35% |
| Secondary | 31.99% | 34.69% | 33.32% | 31.74% | 37.97% | 30.29% | 33.03% | 35.44% | 31.53% | 24.21% | 27.74% | 48.05% |
| Higher | 12.72% | 30.99% | 56.29% | 16.47% | 38.11% | 45.42% | 20.35% | 37.98% | 41.68% | 15.81% | 28.14% | 56.05% |
| p-value | <0.001 | | | <0.001 | | | <0.001 | | | <0.001 | | |
| **Education gap** | | | | | | | | | | | | |
| No difference | 32.91% | 32.82% | 34.26% | 32.57% | 37.00% | 30.43% | 33.74% | 34.14% | 32.12% | 23.23% | 26.02% | 50.75% |
| Higher than partner | 36.17% | 34.09% | 29.73% | 32.50% | 36.06% | 31.44% | 34.79% | 35.58% | 29.63% | 25.93% | 27.13% | 46.94% |
| Lower than partner | 33.02% | 33.73% | 33.25% | 36.02% | 34.88% | 29.10% | 33.03% | 33.75% | 33.22% | 23.93% | 26.20% | 49.87% |
| p-value | 0.006 | | | 0.002 | | | 0.011 | | | <0.001 | | |
| **Number of living children** | | | | | | | | | | | | |
| No child | 57.36% | 18.27% | 24.37% | 48.58% | 44.75% | 6.67% | 48.06% | 44.25% | 7.68% | 34.59% | 31.97% | 33.44% |
| 1–2 | 31.18% | 34.75% | 34.07% | 30.59% | 35.42% | 33.99% | 32.21% | 33.60% | 34.18% | 22.76% | 26.48% | 50.76% |
| 3–4 | 33.08% | 33.05% | 33.87% | 31.75% | 35.41% | 32.84% | 30.92% | 33.59% | 35.48% | 22.61% | 24.56% | 52.83% |
| 4+ | 40.81% | 31.34% | 27.85% | 37.51% | 34.58% | 27.91% | 38.31% | 31.00% | 30.69% | 26.23% | 24.92% | 48.85% |
| p-value | <0.001 | | | <0.000 | | | <0.001 | | | <0.001 | | |
| **Difference between sons and daughters** | | | | | | | | | | | | |
| No difference | 33.72% | 33.30% | 32.99% | 35.76% | 38.18% | 26.06% | 37.10% | 36.06% | 26.84% | 25.28% | 27.66% | 47.06% |
| More sons | 33.75% | 33.25% | 33.01% | 31.97% | 35.63% | 32.40% | 32.76% | 33.18% | 34.06% | 23.47% | 26.08% | 50.45% |
| More daughters | 33.52% | 33.65% | 32.83% | 32.59% | 35.26% | 32.15% | 32.08% | 34.30% | 33.62% | 23.80% | 25.45% | 50.75% |
| p-value | 0.998 | | | <0.001 | | | <0.001 | | | <0.001 | | |
| **Number of household members** | | | | | | | | | | | | |
| 1–4 | 28.64% | 34.25% | 37.11% | 28.47% | 36.53% | 35.00% | 29.71% | 34.46% | 35.84% | 20.58% | 24.59% | 54.84% |

*(Continued)*

**Table 2.** (*Continued*)

| Covariates | BDHS 2007 | | | BDHS 2011 | | | BDHS 2014 | | | BDHS 2017–18 | | |
|---|---|---|---|---|---|---|---|---|---|---|---|---|
| | Low | Medium | High | Low | Medium | High | Low | Medium | High | Low | Medium | High |
| 5–7 | 34.39% | 33.35% | 32.26% | 34.80% | 35.51% | 29.69% | 34.99% | 34.02% | 30.98% | 25.58% | 26.69% | 47.73% |
| 7+ | 40.02% | 32.03% | 27.95% | 40.49% | 37.65% | 21.86% | 41.85% | 35.43% | 22.71% | 30.24% | 30.52% | 39.24% |
| p-value | <0.001 | | | <0.001 | | | <0.001 | | | <0.001 | | |
| **Religion** | | | | | | | | | | | | |
| Non-Muslim | 32.03% | 37.17% | 30.80% | 28.19% | 37.38% | 34.43% | 29.82% | 33.04% | 37.14% | 18.87% | 27.58% | 53.55% |
| Muslim | 33.86% | 32.94% | 33.19% | 34.01% | 36.13% | 29.86% | 34.28% | 34.57% | 31.15% | 24.73% | 26.24% | 49.03% |
| p-value | 0.039 | | | <0.001 | | | <0.001 | | | <0.001 | | |
| **Media exposure** | | | | | | | | | | | | |
| No | 40.64% | 32.12% | 27.23% | 40.00% | 33.11% | 26.90% | 40.80% | 31.59% | 27.60% | 28.93% | 24.97% | 46.11% |
| Yes | 29.85% | 34.05% | 36.10% | 30.02% | 37.86% | 32.12% | 29.85% | 36.05% | 34.10% | 21.63% | 27.12% | 51.26% |
| p-value | <0.001 | | | <0.001 | | | <0.001 | | | <0.001 | | |
| **NGO membership** | | | | | | | | | | | | |
| No | 33.52% | 32.51% | 33.97% | 34.20% | 36.19% | 29.61% | 34.90% | 33.95% | 31.15% | | | |
| Yes | 33.94% | 34.72% | 31.34% | 31.43% | 36.46% | 32.11% | 31.81% | 35.33% | 32.85% | | | |
| p-value | 0.022 | | | 0.001 | | | <0.001 | | | - | | |
| **Wealth index** | | | | | | | | | | | | |
| Poorest | 37.37% | 33.88% | 28.75% | 39.35% | 33.77% | 26.88% | 42.04% | 32.10% | 25.85% | 28.40% | 24.25% | 47.34% |
| Poorer | 40.9% | 32.07% | 27.04% | 39.84% | 33.63% | 26.54% | 38.76% | 33.16% | 28.08% | 27.48% | 25.77% | 46.75% |
| Middle | 39.58% | 32.03% | 28.40% | 37.54% | 35.50% | 26.97% | 35.30% | 35.12% | 29.59% | 25.39% | 27.88% | 46.73% |
| Richer | 33.22% | 35.27% | 31.50% | 31.70% | 39.23% | 29.07% | 32.41% | 35.97% | 31.62% | 23.48% | 28.61% | 47.92% |
| Richest | 22.34% | 33.46% | 44.20% | 22.05% | 38.12% | 39.83% | 22.88% | 35.27% | 41.85% | 17.28% | 25.33% | 57.39% |
| p-value | <0.001 | | | <0.001 | | | <0.001 | | | <0.001 | | |
| **Working status** | | | | | | | | | | | | |
| No | 36.12% | 32.35% | 31.53% | 34.76% | 36.43% | 28.81% | 35.69% | 33.86% | 30.45% | 25.38% | 28.91% | 45.71% |
| Yes | 27.51% | 35.93% | 36.56% | 22.75% | 35.12% | 42.14% | 29.66% | 35.70% | 34.64% | 22.77% | 23.55% | 53.68% |
| p-value | <0.001 | | | <0.001 | | | <0.001 | | | <0.001 | | |
| **Husband's occupation** | | | | | | | | | | | | |
| Farmer | 37.93% | 34.99% | 27.09% | 37.56% | 35.13% | 27.32% | 38.24% | 33.43% | 28.32% | 26.07% | 26.42% | 47.51% |
| Labor | 34.16% | 32.33% | 33.51% | 33.42% | 37.19% | 29.39% | 33.76% | 34.47% | 31.77% | 24.59% | 26.44% | 48.97% |
| Service | 13.80% | 29.49% | 56.71% | 19.35% | 37.30% | 43.35% | 19.11% | 35.97% | 44.92% | 13.66% | 24.56% | 61.78% |
| Large business | 26.59% | 37.19% | 36.22% | 27.46% | 36.90% | 35.64% | 26.95% | 32.93% | 40.12% | 17.75% | 29.65% | 52.61% |
| Small business | 35.20% | 34.19% | 30.60% | 33.26% | 35.71% | 31.02% | 33.53% | 35.71% | 30.76% | 24.96% | 26.42% | 48.62% |
| Unemployed | 35.17% | 28.28% | 36.55% | 38.01% | 35.59% | 26.39% | 45.28% | 33.02% | 21.70% | 25.00% | 0.0% | 75.00% |
| Others | 30.61% | 27.55% | 41.84% | 42.19% | 29.69% | 28.13% | 38.36% | 31.48% | 30.16% | 24.43% | 25.80% | 49.77% |
| p-value | <0.001 | | | <0.001 | | | <0.001 | | | <0.001 | | |
| **Relationship with household head** | | | | | | | | | | | | |
| Head | 21.85% | 28.15% | 50.00% | 20.21% | 30.05% | 49.73% | 23.00% | 29.07% | 47.93% | 16.76% | 19.72% | 63.53% |
| Wife | 32.77% | 34.05% | 33.18% | 30.93% | 35.92% | 33.14% | 31.93% | 34.06% | 34.01% | 21.73% | 24.91% | 53.36% |
| Daughter | 33.68% | 31.95% | 34.37% | 36.64% | 39.31% | 24.05% | 35.97% | 38.52% | 25.51% | 22.85% | 31.25% | 45.90% |
| Daughter-in-law | 45.21% | 33.33% | 21.45% | 48.95% | 37.35% | 13.70% | 47.81% | 35.63% | 16.56% | 41.76% | 33.08% | 25.16% |
| Grand-daughter | 25.00% | 50.00% | 25.00% | 71.43% | 23.81% | 4.76% | 29.17% | 54.17% | 16.67% | 18.42% | 42.11% | 39.47% |
| Mother | 50.00% | 15.00% | 35.00% | 38.94% | 36.28% | 24.78% | 35.00% | 37.00% | 28.00% | 29.52% | 29.52% | 40.95% |
| Mother-in-law | 27.27% | 36.36% | 36.36% | 11.54% | 65.38% | 23.08% | 34.78% | 21.74% | 43.48% | 28.57% | 7.14% | 64.29% |
| Sister | 39.68% | 28.57% | 31.75% | 38.12% | 39.01% | 22.87% | 31.58% | 39.47% | 28.95% | 17.86% | 31.25% | 50.89% |
| Other relative | 36.08% | 34.12% | 29.80% | 39.22% | 39.75% | 21.02% | 41.04% | 36.85% | 22.11% | 29.33% | 31.73% | 38.94% |
| Adopted child | 75.00% | 25.00% | 0.0% | 25.00% | 25.00% | 50.00% | 40.00% | 20.00% | 40.00% | 14.29% | 71.43% | 14.29% |

(*Continued*)

**Table 2.** (Continued)

| Covariates | BDHS 2007 | | | BDHS 2011 | | | BDHS 2014 | | | BDHS 2017–18 | | |
|---|---|---|---|---|---|---|---|---|---|---|---|---|
| | Low | Medium | High | Low | Medium | High | Low | Medium | High | Low | Medium | High |
| Not related | 25.0% | 50.00% | 25.00% | 40.00% | 30.00% | 30.00% | 30.77% | 46.15% | 23.08% | 14.29% | 28.57% | 57.14% |
| p-value | <0.001 | | | <0.001 | | | <0.001 | | | <0.001 | | |
| **Sex of household head** | | | | | | | | | | | | |
| Male | 34.20% | 33.84% | 31.96% | 33.79% | 36.55% | 29.66% | 34.44% | 34.78% | 30.78% | 24.55% | 26.67% | 48.79% |
| Female | 28.14% | 28.27% | 43.59% | 27.74% | 32.71% | 39.55% | 27.87% | 30.74% | 41.39% | 21.08% | 24.16% | 54.75% |
| p-value | <0.001 | | | <0.001 | | | <0.001 | | | <0.001 | | |

p-value[A]: p-value of ANOVA F-test.

with household head. In brief, women living in Sylhet division had lower odds of gaining empowerment as compared to Dhaka dwellers women ($OR_{2007}$: 0.641, $OR_{2011}$: 0.670, $OR_{2011}$: 0.643, $OR_{2017-18}$: 0.765). Women from rural areas had significantly lesser influence (12.6%, 16.2%, 7.9%, and 24.2% lower odds, respectively) in receiving empowerment than those who were from urban areas. There were significant differences in the level of women's empowerment among secondary and higher educated women than uneducated. Women who completed secondary education in 2007, 2011, 2014, and 2017–18, respectively have 14.4%, 31.8%, 24.6%, and 39.6% higher odds of having empowerment compared to those who were uneducated. Further, higher educated women in the four BDHS waves, respectively were 2.083, 2.201, 1.794 and 1.935 times as likely to have empowerment compared to those who had no education. Number of living children has been seen as one of the most important covariates in this study with p-value <0.001. The likelihood of having 3–4 children than those couples who had no children from 2007 to 2017–18 BDHS, respectively was 1.978, 3.191, 3.119, and 2.095 times as likely to have empowerment. Similarly, couples with 1–2 children and 4+ children, respectively were more likely to have empowerment than couples with no children. In the present study, women who belong to small size families have a better chance to establish empowerment. Similarly, women from richest families, and are exposed to media have higher odds of being empowered. As expected, the employment status of women in Bangladesh was strongly associated with their empowerment. More precisely, the likelihood of being empowered was 1.271, 1.529, 1.161, and 1.283 times as likely for employed women for the study years, respectively, compared to homemakers. Comparing women who themselves are head of the household with those who have other relationship (i.e., wife, daughter, daughter-in-law, sister etc.) with household head were less likely to have empowerment in household decision-making as well as attitudes toward wife beating.

## Discussion

Women's empowerment is a pivotal determinant of sustainable development, attracting global attention due to its impact on societal power dynamics [28,35,36]. Numerous studies have explored empowerment through decision-making power or attitudes toward wife beating, and our study contributes by focusing on influential determinants in Bangladesh over a decade [17,18,23,24,26–30,36–44]. Our analysis, rooted in previous research, identified key factors associated with women's empowerment. Bivariate analysis highlighted strong associations between several covariates and empowerment, emphasizing the role of marriage age, education, media exposure, employment, and socioeconomic status [14,27,28]. Adjusted odds ratios further elucidated these associations, emphasizing their significance in shaping women's empowerment.

**Table 3. Association between different background characteristics with women's empowerment obtained from adjusted proportional odds model (POM).**

| Background characteristics | BDHS 2007 | | BDHS 2011 | | BDHS 2014 | | BDHS 2017–18 | |
|---|---|---|---|---|---|---|---|---|
| | OR | p-value | OR | p-value | OR | p-value | OR | p-value |
| **Division** | | | | | | | | |
| Dhaka | 1 | | 1 | | 1 | | 1 | |
| Chittagong | 0.658 | <0.001 | 0.730 | <0.001 | 0.772 | <0.001 | 0.937 | 0.224 |
| Barisal | 0.597 | <0.001 | 1.014 | 0.797 | 0.593 | <0.001 | 0.770 | <0.001 |
| Khulna | 0.999 | 0.994 | 1.053 | 0.319 | 0.658 | <0.001 | 0.908 | 0.077 |
| Mymensingh | - | - | - | - | - | - | 1.475 | <0.001 |
| Rajshahi | 0.998 | 0.973 | 0.605 | <0.001 | 0.693 | <0.001 | 0.949 | 0.349 |
| Rangpur | - | - | 1.336 | <0.001 | 0.900 | 0.051 | 0.967 | 0.554 |
| Sylhet | 0.641 | <0.001 | 0.670 | <0.001 | 0.643 | <0.001 | 0.765 | <0.001 |
| **Place of residence** | | | | | | | | |
| Urban | 1 | | 1 | | 1 | | 1 | |
| Rural | 0.874 | 0.006 | 0.838 | <0.001 | 0.921 | 0.022 | 0.758 | <0.001 |
| **Age at first marriage** | | | | | | | | |
| Age<18 | 1 | | 1 | | 1 | | 1 | |
| Age≥18 | 1.165 | 0.008 | 1.059 | 0.147 | 1.126 | 0.002 | 1.090 | 0.017 |
| **Spousal age gap** | 0.997 | 0.300 | 1.005 | 0.049 | 0.992 | 0.003 | 1.001 | 0.771 |
| **Education level** | | | | | | | | |
| No education | 1 | | 1 | | 1 | | 1 | |
| Primary | 1.033 | 0.557 | 1.055 | 0.225 | 1.033 | 0.474 | 1.166 | 0.001 |
| Secondary | 1.144 | 0.040 | 1.318 | <0.001 | 1.246 | <0.001 | 1.396 | <0.001 |
| Higher | 2.083 | <0.001 | 2.201 | <0.001 | 1.794 | <0.001 | 1.935 | <0.001 |
| **Education gap** | | | | | | | | |
| No difference | 1 | | 1 | | 1 | | 1 | |
| Higher than partner | 0.915 | 0.110 | 0.954 | 0.244 | 0.995 | 0.896 | 0.883 | <0.001 |
| Lower than partner | 1.009 | 0.864 | 1.065 | 0.097 | 1.044 | 0.274 | 1.020 | 0.609 |
| **Number of living children** | | | | | | | | |
| No child | 1 | | 1 | | 1 | | 1 | |
| 1–2 | 1.817 | <0.001 | 2.853 | <0.001 | 2.503 | <0.001 | 1.809 | <0.001 |
| 3–4 | 1.978 | <0.001 | 3.191 | <0.001 | 3.119 | <0.001 | 2.095 | <0.001 |
| 4+ | 1.815 | 0.001 | 3.154 | <0.001 | 2.950 | <0.001 | 2.100 | <0.001 |
| **Difference between sons and daughter** | | | | | | | | |
| No difference | 1 | | 1 | | 1 | | 1 | |
| More sons | 0.970 | 0.583 | 0.869 | 0.001 | 0.967 | 0.423 | 0.892 | 0.004 |
| More daughters | 0.962 | 0.518 | 0.902 | 0.018 | 0.968 | 0.442 | 0.890 | 0.005 |
| **Number of household members** | | | | | | | | |
| 1–4 | 1 | | 1 | | 1 | | 1 | |
| 5–7 | 0.876 | 0.014 | 0.838 | <0.001 | 0.848 | <0.001 | 0.884 | 0.001 |
| 7+ | 0.805 | 0.003 | 0.746 | <0.001 | 0.700 | <0.001 | 0.813 | <0.001 |
| **Wealth index** | | | | | | | | |
| Middle | 1 | | 1 | | 1 | | 1 | |
| Poorest | 1.056 | 0.460 | 0.931 | 0.215 | 0.799 | <0.001 | 1.013 | 0.937 |
| Poorer | 0.952 | 0.436 | 0.940 | 0.179 | 0.918 | 0.079 | 0.996 | 0.796 |
| Richer | 1.110 | 0.111 | 1.122 | 0.016 | 1.026 | 0.581 | 0.980 | 0.653 |
| Richest | 1.446 | <0.001 | 1.512 | <0.001 | 1.377 | <0.001 | 1.26 | <0.001 |
| **Religion** | | | | | | | | |
| Non-Muslim | 1 | | 1 | | 1 | | 1 | |

*(Continued)*

**Table 3.** (Continued)

| Background characteristics | BDHS 2007 | | BDHS 2011 | | BDHS 2014 | | BDHS 2017–18 | |
|---|---|---|---|---|---|---|---|---|
| | OR | p-value | OR | p-value | OR | p-value | OR | p-value |
| Muslim | 0.936 | 0.323 | 0.845 | <0.001 | 0.770 | <0.001 | 0.759 | <0.001 |
| **Media exposure** | | | | | | | | |
| No | 1 | | 1 | | 1 | | 1 | |
| Yes | 1.137 | 0.010 | 1.104 | 0.009 | 1.103 | 0.011 | 1.115 | 0.001 |
| **NGO membership** | | | | | | | | |
| No | 1 | | 1 | | 1 | | - | - |
| Yes | 0.961 | 0.359 | 1.073 | 0.037 | 1.082 | 0.018 | - | - |
| **Working status** | | | | | | | | |
| No | 1 | | 1 | | 1 | | 1 | |
| Yes | 1.271 | <0.001 | 1.529 | <0.001 | 1.161 | <0.001 | 1.283 | <0.001 |
| **Husband's occupation** | | | | | | | | |
| Farmer | 1 | | 1 | | 1 | | 1 | |
| Labor | 1.060 | 0.273 | 0.948 | 0.189 | 1.025 | 0.546 | 1.013 | 0.739 |
| Service | 1.543 | <0.001 | 1.088 | 0.254 | 1.258 | 0.002 | 1.184 | 0.026 |
| Large business | 1.037 | 0.710 | 0.973 | 0.704 | 1.115 | 0.243 | 1.001 | 0.990 |
| Small business | 0.972 | 0.654 | 0.928 | 0.122 | 0.937 | 0.157 | 0.926 | 0.090 |
| Unemployed | 1.224 | 0.219 | 0.890 | 0.246 | 0.791 | 0.210 | 2.972 | 0.364 |
| Others | 1.428 | 0.078 | 0.928 | 0.764 | 0.905 | 0.336 | 1.043 | 0.666 |
| **Relationship with household head** | | | | | | | | |
| Head | 1 | | 1 | | 1 | | 1 | |
| Wife | 0.501 | <0.001 | 0.490 | <0.001 | 0.642 | <0.001 | 0.681 | 0.001 |
| Daughter | 0.524 | <0.001 | 0.443 | <0.001 | 0.631 | <0.001 | 0.634 | <0.001 |
| Daughter-in-law | 0.301 | <0.001 | 0.249 | <0.001 | 0.370 | <0.001 | 0.265 | <0.001 |
| Grand-daughter | 0.505 | 0.443 | 0.125 | <0.001 | 0.689 | 0.315 | 0.591 | 0.080 |
| Mother | 0.306 | 0.001 | 0.383 | <0.001 | 0.604 | 0.020 | 0.517 | 0.001 |
| Mother-in-law | 0.644 | 0.298 | 0.589 | 0.136 | 0.727 | 0.448 | 0.861 | 0.718 |
| Sister | 0.438 | <0.001 | 0.411 | <0.001 | 0.773 | 0.142 | 0.764 | 0.088 |
| Other relative | 0.418 | <0.001 | 0.340 | <0.001 | 0.491 | <0.001 | 0.456 | <0.001 |
| Adopted child | 0.086 | 0.034 | 0.893 | 0.910 | 1.140 | 0.877 | 0.284 | 0.041 |
| Not related | 0.380 | 0.273 | 0.384 | 0.117 | 0.457 | 0.132 | 0.627 | 0.528 |
| **Sex of household head** | | | | | | | | |
| Male | 1 | | 1 | | 1 | | 1 | |
| Female | 0.966 | 0.795 | 0.953 | 0.632 | 1.032 | 0.749 | 1.032 | 0.683 |

Urban-rural disparities emerged as a crucial factor, with urban women experiencing greater empowerment, particularly in Dhaka. Education played a pivotal role, corroborating findings from prior studies, showcasing the link between education, decision-making ability, and resistance against intimate partner violence [28,36,45–47]. However, the unexpected negative association observed for highly educated wives warrants further investigation. Family dynamics, including the number of children and their gender distribution, revealed nuanced insights. Larger families were associated with reduced empowerment, while the impact of sons versus daughters on empowerment exhibited variations across study years. These findings challenge conventional expectations, suggesting the need for a nuanced understanding of family dynamics in the context of women's empowerment.

Wealth quintiles highlighted socioeconomic disparities, with richer women consistently exhibiting higher odds of empowerment. Religious affiliation, specifically Muslim identity,

emerged as a significant factor impacting women's empowerment, echoing the influence of religious beliefs on societal norms and behaviors [38,48,49]. Media exposure and employment status corroborated past research, showing positive associations with empowerment [46], while NGO membership emerged as a contributing factor in specific survey years.

Spousal occupation also played a role, emphasizing the financial stability's impact on women's empowerment [50]. The role of women as household heads showcased a positive association with empowerment, but the low prevalence of female household heads underscored the patriarchal nature of Bangladeshi society. Additionally, regional disparities and variations over the study period highlight the need for targeted interventions based on geographical contexts.

## Conclusion

The findings of the study demonstrate that the situation of women's empowerment in Bangladesh has improved over the survey years, as evidenced by the indicators: household decision-making and attitudes toward violence. Our study confirmed that women's empowerment is significantly associated with factors such as division, place of residence, education level, number of living children, wealth index, media exposure, working status, and relationship with household head. Additionally, this study also provides evidence that urban dwellers women, secondary educated women, women with 1–2 children, exposed to mass media, employed women reported having more empowerment than the others over the ten-year span of time. Ensuring further steps to open door to women's education, raising awareness among women, and providing more facilities can accelerate women's empowerment rapidly and in a better way.

## Limitations

Our study is not beyond limitations. Firstly, this study used data extracted from BDHS having in responses only from women which introduced biases to some extent. Secondly, women's empowerment was defined in our analysis using only two indicators (household decision-making and attitudes toward wife beating) but this could be more strongly defined using more indicators like economic decision-making, access to healthcare, physical mobility, the decision on family planning, assets ownership, etc. Due to lack of data, some important covariates were not considered in this study.

## Acknowledgments

The Demographic and Health Survey (DHS), which carried out a nationwide survey and made its data openly accessible, is appreciated by the authors. We would like to express our gratitude and homage to honorable Professor the late Dr. Md. Taslim Sazzad Mallick, Department of Statistics, University of Dhaka. Additionally, we would like to express our gratitude to the academic editor and the anonymous reviewers for their contributions that enhanced the manuscript's quality and coherence.

## Author Contributions

**Conceptualization:** Sahera Akter, Md. Solayman Hosen, Md. Shehab Khan, Bikash Pal.

**Data curation:** Sahera Akter, Md. Solayman Hosen, Md. Shehab Khan.

**Formal analysis:** Sahera Akter, Md. Solayman Hosen, Md. Shehab Khan.

**Methodology:** Sahera Akter.

**Software:** Sahera Akter, Md. Solayman Hosen, Md. Shehab Khan.

**Supervision:** Bikash Pal.

**Writing – original draft:** Sahera Akter, Md. Solayman Hosen, Md. Shehab Khan.

**Writing – review & editing:** Bikash Pal.

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
