## [Decision Letter · Decision Letter 0]

2 Jan 2024

PONE-D-23-17327Assessing the pattern of key factors on women’s empowerment in Bangladesh: evidence from Bangladesh Demographic and Health Survey, 2007 to 2017-18PLOS ONE

Dear Dr. Akter,

Thank you for submitting your manuscript to PLOS ONE. After careful consideration, we feel that it has merit but does not fully meet PLOS ONE’s publication criteria as it currently stands. Therefore, we invite you to submit a revised version of the manuscript that addresses the points raised during the review process.

We look forward to receiving your revised manuscript.

Kind regards,

Mohammad Nayeem Hasan

Academic Editor

PLOS ONE

Journal Requirements:

Additional Editor Comments:

Manuscript ID PONE-D-23-17327 entitled "Assessing the pattern of key factors on women’s empowerment in Bangladesh: evidence from Bangladesh Demographic and Health Survey, 2007 to 2017-18" which you submitted to the PLOS ONE, has been reviewed and will be reconsidered for publication after the completion of the major revisions as noted. The comments of two reviewers are included at the bottom of this letter. My attempts at obtaining some other reviewers to improve your paper were not successful. Rather than postpone the review process further, I have decided to serve as the 3rd reviewer to increase the quality of this paper.

The paper investigated significant factors of women’s empowerment and at the same time, assessed the pattern of potential factors affecting women’s empowerment in Bangladesh using four cross-sectional waves of Bangladesh Demographic and Health Surveys (2007, 2011, 32 2014, and 2017-18) data. Despite being a significant concern for Bangladesh, this topic has already received considerable attention from numerous earlier and recently published studies.

https://www.tandfonline.com/doi/abs/10.1080/00036846.2015.1051657

https://bmjopen.bmj.com/content/11/8/e049167.abstract

https://link.springer.com/article/10.1186/s12905-020-00952-4

https://www.emerald.com/insight/content/doi/10.1108/JHR-11-2020-0559/full/html

The purpose of the study has also been used in the past to explore comparable issues utilizing earlier data on Bangladesh. As a result, the paper's main original addition appears to be its comparison between different survey data, and its conclusions largely—and not surprisingly—reflect those of previous research. The paper needs to be thoroughly proofread by a professional or expert. It is not written to a high degree for academic writing, and there are several grammatical/spelling errors, which frequently make claims and arguments less clear. Additionally, the authors ought to eliminate any similarities from this work. Numerous times, the way that statistical results are interpreted is also wrong. The introduction fails to effectively justify the need for studying the issue, tables and results are not focused on the title, many unusual tables and results are included in the main manuscript, and the paper's findings by themselves are insufficient to support the conclusions.

Below are some comments with more information:

Abstract

The methods, results, and conclusion need to be re-written. Please check previously published papers from PLOS ONE.

Introduction

Overall, the introduction is so long. Please reduce some text or concise it. In many places of the introduction, citation is needed. Please check and provide the appropriate citation.

Lines 117-118: “At the same time, an evaluation of women's empowerment in Bangladesh over the past 10 years, from 2007 to 2017-18 has also been observed here.” As you compared 4 surveys and it is not time series data, so, 10 years explanation in the title and objective is not acceptable.

Please specify the research gap. Why this study is necessary and so on. I didn’t find it in your introduction.

Data and Methodology

Data

Table 1 is not needed here as all procedures are shared on the BDHS report. Those who want to check that they will check in the reports. Rather, it is better to show a flow chart of how you reached your final sample (every step of inclusion-exclusion)

Outcome variable

Lines 163-165: “We proposed a total of nine questions which were classified into two broad dimensions; household decision-making and attitudes toward wife beating”. Is it the authors' proposal or followed some published papers, please re-check it.

Lines 179-183: “For creating the women’s empowerment index (WEI), principal component analysis (PCA) has been used with all nine indicators (4 decisions and 5 reasons), where the first principal component was regarded as the women’s empowerment score (WES). The WES was further broken down into 3 quantiles; labeled low, middle, and high for domains below the first, in between the first and second, and above the second quantile, respectively.”. Please explain more about this procedure. How many variables remained after PCA, how did the authors do that, and what’s the methodology of doing it? How did you categorize that on what basis, etc.?

Lines 183-186: The author should write “Finally, our preferred outcome metric is the score index with three ordered categories (i.e., low, medium, and high), where these categories indicate order-wise how empowered a woman is (i.e., low means women have low empowerment)”. Only citation is not appropriate here. As you did your own PCA, you should have a different score from the others. So, please explain your score to categorize them.

Covariates

Lines 191-195: “Based on some previous works of literature, the covariates included in this study are age at first marriage, spousal age gap, respondent’s education level, education gap, respondent’s current working status, number of living children, religion, number of household members, division, place of residence, media exposure, NGO membership, wealth index, husband’s occupation, relationship with household head, and sex of household head.” Please cite all of the previous works after those lines.

Table 2 is not enough to explain the details about covariates. Please cite the appropriate BDHS methodological reports to explain the categories. Or if you follow other reports to categorize them, you can also make them in citation. To remove any confusion, follow this: https://bmcpublichealth.biomedcentral.com/articles/10.1186/s12889-023-15617-8

Statistical analysis

Please provide some model evaluation techniques (AIC, BIC, AUROC, and others…..) to justify that you used an appropriate model to explain your results. In addition, explain about level of significance in statistical analysis.

Ordinal logistic regression model

No need to explain in detail the ordinal logistic regression model, as it is a common method to us. The authors can explain the application of it.

Please explain why the authors used two statistical software, if you just need two software, please explain which statistical analysis performs with which software.

Results

There are some issues with the representations and interpretations of the results.

Univariate analysis

Table 3 is not appropriate in this paper. Please represent Table 3 with the outcome variable and show the P-value in this same table. We need to see the change in women's empowerment situation overtime over the socio-demographic variables. The explanation of Table 3 is too large. Please reduce it by a significant variable. Explain only significant variables by performing a chi-square test on it.

Bivariate analysis

Lines 280-283: “In the bivariate analysis, an attempt has been made to out find the significant exposure factors to women’s empowerment. To evaluate how much a woman's background characteristics influence her level of domestic empowerment in Bangladesh, Pearson Chi-square test was applied.” Please remove these lines from here, because it is a methodological sentence.

Merge Table 4 in Table 3. No need to show a chi-square test value. Show only the P-value (from the chi-square test) for significant tests.

Multivariate analysis

Instead of “multivariate analysis”, please explain it as multivariable analysis.

The authors said before in lines 207-208; “The covariates that were significantly associated with the outcome variable, have been included in the regression model”. However in regression analysis, the authors include all variables.

The representation of Table 5 needs to be followed in a standard format. Please follow this paper’s Table 3 and Table 4 reporting style: https://journals.plos.org/plosone/article?id=10.1371/journal.pone.0242864

Use <0.001 when it is less than 0.001/<0.01. Explain the P-value as 3 decimal points.

It also seems that the author didn’t maintain the serial of explanation of Table 5 or vice versa. Please, to catch the reader’s interest please keep the flow similar to the table and interpretation.

No need to explain whether it is a 1% or 5% significant level. You should follow only one level of significance when you explain.

The interpretation of table 5 is also very long. Please explain only key significant variables and key findings.

Discussion

The discussion part should be rewritten and should focus on outcome variables. It will be better to discuss the information of supplementary tables and be more specific on the outcome variable. The authors discuss the reason for each association in the discussion part.

Major findings of this study

All explanations of figures need to be replaced in the results section. The quality and the representation of the figure are not good.

Conclusion

The conclusion needs to be more specific and highlight key findings. This should support the result. They are suggested to provide less discussion in the conclusion part. I will suggest providing no citation in the conclusion part.

You can follow the below publication with BDHS/MICS data, try to represent in these styles:

https://bmcpublichealth.biomedcentral.com/articles/10.1186/s12889-023-15617-8

https://aidsrestherapy.biomedcentral.com/articles/10.1186/s12981-022-00495-8

https://www.frontiersin.org/articles/10.3389/fpubh.2022.985445/full

https://journals.plos.org/plosone/article?id=10.1371/journal.pone.0242864

These suggestions aim to enhance the study's coherence, precision, and informative value, ensuring it provides valuable insights.

Reviewers' comments:

**Comments to the Author**

1. Is the manuscript technically sound, and do the data support the conclusions?

Reviewer #1: Yes

Reviewer #2: Yes

2. Has the statistical analysis been performed appropriately and rigorously? 

Reviewer #1: N/A

Reviewer #2: Yes

3. Have the authors made all data underlying the findings in their manuscript fully available?

Reviewer #1: Yes

Reviewer #2: Yes

4. Is the manuscript presented in an intelligible fashion and written in standard English?

Reviewer #1: Yes

Reviewer #2: Yes

5. Review Comments to the Author

Reviewer #1: Thanks for inviting me to review this manuscript. Over all is sounds nicely written, constructed, based on many data, with right methodology, constructive discussion and relevant references.and contribute to the literature. I recommend for publication without any major revisions.

Reviewer #2: Dear authors,

Your manuscript is technically sound, and the data supported the conclusions.

Empowering women in Bangladesh is not only a moral obligation, but also a crucial step towards creating a more just and prosperous society. Your study analyzed data from four cross-sectional waves of the Bangladesh Demographic and Health Surveys to identify the factors that contribute to women's empowerment. And found that education, employment, higher wealth index, media exposure, NGO membership, and urban residency all significantly increased the likelihood of women experiencing empowerment.

The statistical analysis had been performed appropriately.

The principal component analysis (PCA) was employed to create women's empowerment index. The Chi-square test was used to assess the unadjusted association between the selected covariates and women's empowerment. And the proportional odds model (POM) was applied to determine the adjusted association of the selected covariates with women's empowerment.

All data underlying the findings in your manuscript are fully available for readers.

Finally, the manuscript is presented in an intelligible fashion and written in standard English.

Sincerely,

Beisan Ali

6. PLOS authors have the option to publish the peer review history of their article (what does this mean?). If published, this will include your full peer review and any attached files.

Reviewer #1: No

Reviewer #2: **Yes: **Beisan A. Mohammad

---

## [Author Response · Author response to Decision Letter 0]

5 Feb 2024

The Academic Editor and the reviewers: We have incorporated all of your suggestions into my revision. They were very helpful. Thank you.

---

## [Decision Letter · Decision Letter 1]

18 Mar 2024

Assessing the pattern of key factors on women’s empowerment in Bangladesh: evidence from Bangladesh Demographic and Health Survey, 2007 to 2017-18

PONE-D-23-17327R1

Dear Dr. Sahera Akter,

We’re pleased to inform you that your manuscript has been judged scientifically suitable for publication and will be formally accepted for publication once it meets all outstanding technical requirements.

Kind regards,

Mohammad Nayeem Hasan

Academic Editor

PLOS ONE

Additional Editor Comments (optional):

Reviewers' comments:

Reviewer's Responses to Questions

**Comments to the Author**

1. If the authors have adequately addressed your comments raised in a previous round of review and you feel that this manuscript is now acceptable for publication, you may indicate that here to bypass the “Comments to the Author” section, enter your conflict of interest statement in the “Confidential to Editor” section, and submit your "Accept" recommendation.

Reviewer #3: All comments have been addressed

Reviewer #4: All comments have been addressed

Reviewer #5: All comments have been addressed

2. Is the manuscript technically sound, and do the data support the conclusions?

Reviewer #3: Yes

Reviewer #4: Yes

Reviewer #5: Yes

3. Has the statistical analysis been performed appropriately and rigorously? 

Reviewer #3: Yes

Reviewer #4: Yes

Reviewer #5: I Don't Know

4. Have the authors made all data underlying the findings in their manuscript fully available?

Reviewer #3: Yes

Reviewer #4: Yes

Reviewer #5: Yes

5. Is the manuscript presented in an intelligible fashion and written in standard English?

Reviewer #3: Yes

Reviewer #4: Yes

Reviewer #5: Yes

6. Review Comments to the Author

Reviewer #3: 1. Although the authors have addressed all of the comments, there are certain instances of grammatical and spelling errors in the manuscript. As such, the authors are advised to proofread the manuscript and make relevant corrections.

2. Abstract, results section, lines (51-54): “For instance, women who completed secondary education in 2007, 2011, 2014, and 2017-18, respectively have 14.4%, 31.8%, 24.6%, and 39.6% higher odds of having empowerment compared to those who were uneducated.” The authors should consider removing the aforementioned lines from the abstract as these are redundant.

3. In the limitations section, the authors mention that due to insufficient data, certain covariates were not considered. Could the authors briefly mention those covariates, highlighting the need for further research in those areas.

Reviewer #4: Dear editor and authors,

The authors addressed the identify factors contributing to women’s empowerment in Bangladesh. They proposed four waves of BDHS data (2007, 2011, 2014, and 2017-18) to assess temporal changes in covariates impacting women’s empowerment. They approach to decision-making and decision-making problems toward wife beating household. Later, they evaluated the comprehensive analysis of evolving factors

Overall, the authors presented detailed information of their motivation and experiment details. The experiment results and discussion support authors' claim. The entire manuscript is well described with scientific soundness on an interval-valued picture fuzzy context. Notations, formulations, tables, materials and methods, and statements of practical results are readable. Discussion, conclusions and limitations are correct.

After checking the revised version of this manuscript. I am satisfied with the efforts the authors made in revising this manuscript. Most concerns raised by reviewers have been addressed, but there are still some errors. The current revised manuscript looks good and could be accepted to publish on “PLOS ONE”. Make the following comments:

• Make the “Abstract” section as one paragraph. Do not use the words “Background”, “Methods”, “Results” and “Conclusion”.

Reviewer #5: If the affiliations are the same for all authors, please remove duplicate entries from the affiliation section.

Rewrite the beginning of the abstract to ensure sentence harmony.

Insert citations at the end of the 1st paragraph, 2nd paragraph, and the last paragraph of the introduction.

As previously suggested, tables have been removed from the manuscript.

The manuscript now includes an explanation about PCA.

Citations have been added to Table 1, improving the referencing within the manuscript.

The results and discussion sections have been updated as per the reviewer's suggestions.

7. PLOS authors have the option to publish the peer review history of their article (what does this mean?). If published, this will include your full peer review and any attached files.

Reviewer #3: No

Reviewer #4: No

Reviewer #5: **Yes: **Dr. Muhammad Asad

---

## [Editor Report · Acceptance letter]

21 Mar 2024

PONE-D-23-17327R1 

PLOS ONE

Dear Dr. Akter, 

I'm pleased to inform you that your manuscript has been deemed suitable for publication in PLOS ONE. Congratulations! Your manuscript is now being handed over to our production team.

Kind regards, 

on behalf of

Dr. Mohammad Nayeem Hasan 

Academic Editor

PLOS ONE